# COVID-19 Related Myocarditis in Adults: A Systematic Review of Case Reports

**DOI:** 10.3390/jcm11195519

**Published:** 2022-09-21

**Authors:** Szymon Urban, Michał Fułek, Mikołaj Błaziak, Gracjan Iwanek, Maksym Jura, Katarzyna Fułek, Mateusz Guzik, Mateusz Garus, Piotr Gajewski, Łukasz Lewandowski, Jan Biegus, Piotr Ponikowski, Przemysław Trzeciak, Agnieszka Tycińska, Robert Zymliński

**Affiliations:** 1Institute of Heart Diseases, Wroclaw Medical University, 50-376 Wroclaw, Poland; 2Department of Internal Medicine, Occupational Diseases, Hypertension and Clinical Oncology, Wroclaw Medical University, 50-376 Wroclaw, Poland; 3Department of Physiology and Pathophysiology, Wroclaw Medical University, 50-376 Wroclaw, Poland; 4Lower Silesian Oncology, Pulmonology and Hematology Center, 50-376 Wroclaw, Poland; 5Department of Medical Biochemistry, Wroclaw Medical University, Chalubinskiego 10, 50-368 Wroclaw, Poland; 63rd Department of Cardiology, Faculty of Medical Sciences in Zabrze, Medical University of Silesia, 40-055 Katowice, Poland; 7Department of Cardiology, Medical University of Bialystok, 15-089 Bialystok, Poland

**Keywords:** COVID-19, myocarditis, cardiology, systematic review, cardiogenic shock

## Abstract

Despite the progress of its management, COVID-19 maintains an ominous condition which constitutes a threat, especially for the susceptible population. The cardiac injury occurs in approximately 30% of COVID-19 infections and is associated with a worse prognosis. The clinical presentation of cardiac involvement can be COVID-19-related myocarditis. Our review aims to summarise current evidence about that complication. The research was registered at PROSPERO (CRD42022338397). We performed a systematic analysis using five different databases, including i.a. MEDLINE. Further, the backward snowballing technique was applied to identify additional papers. Inclusion criteria were: full-text articles in English presenting cases of COVID-19-related myocarditis diagnosed by the ESC criteria and patients over 18 years old. The myocarditis had to occur after the COVID-19 infection, not vaccination. Initially, 1588 papers were screened from the database search, and 1037 papers were revealed in the backward snowballing process. Eventually, 59 articles were included. Data about patients’ sex, age, ethnicity, COVID-19 confirmation technique and vaccination status, reported symptoms, physical condition, laboratory and radiological findings, applied treatment and patient outcome were investigated and summarised. COVID-19-related myocarditis is associated with the risk of sudden worsening of patients’ clinical status, thus, knowledge about its clinical presentation is essential for healthcare workers.

## 1. Introduction

The pandemic of COVID-19 had an immense impact on nearly every aspect of the modern world. Until August 2022 585 million people have been infected, and 6.4 million died [1]. Initially, it was considered a respiratory disease, but it soon occurred to be a multidimensional condition, dangerous for several systems, including cardiovascular [2,3,4]. Cardiac manifestations of COVID are varied and include i.a., arrhythmias, acute coronary syndrome, heart failure decompensation and myocarditis [5]. Myocarditis is an inflammatory disease of the myocardium with vast symptomatology and not entirely ascertained pathogenesis [6]. Since myocarditis can lead to life-threatening conditions and its incidence is interwound with the current most crucial epidemiological problem–COVID-19—w decided to perform a systematic review of the available literature. The aim of our study was to show the up-to-date evidence of epidemiology, clinical course, diagnostics, treatment and prognosis of COVID-related myocarditis.

## 2. Materials and Methods

The study was conducted following PRISMA guidelines [7] and was registered in PROSPERO (no (CRD42022338397).

### 2.1. Search Strategy

A systematic search was performed using EBSCOhost searching engine. MEDLINE, MEDLINE Ultimate, Health Source—Consumer Edition, Health Source: Nursing/Academic Edition, Academic Search Ultimate and ERIC databases were screened. The research included papers published from 1 December 2019 until 1 June 2022. Two researchers (S.U. and M.B.) independently searched databases using the following keywords: (COVID-19 or coronavirus or 2019-ncov or SARS-CoV-2 or CoV-19 or novel coronavirus OR SARS CoV OR SARS-CoV-19) AND (“cardiac damage” OR “cardiac injury” OR “acute cardiac injury” OR “acute myocardial injury” Myocarditis OR perimyocarditis OR endomyocarditis OR endoperimyocarditis). Initially, 1588 papers were identified. After the automatic exclusion of non-English articles and duplicate removal, 695 papers were assessed and eventually, 17 articles were included in the analysis. Studies which one researcher positively qualified were left to the decision of the third investigator (M.F.). Further, reference lists, papers which cited selected articles and two similar systematic reviews [8,9] were screened for relevant articles, which resulted in identifying 1037 and eventually including 57 papers (Figure 1). After the thorough full-text assessment, 59 studies and 70 patients were included in the analysis.

### 2.2. Eligibility Criteria

Inclusion criteria were: full-text peer-reviewed articles in English, case-report formula, age of patient ≥18 years, myocarditis after the COVID infection, laboratory-confirmed COVID infection (including COVID in medical history), diagnosis of myocarditis by ESC criteria [6]. Exclusion criteria were: incident of myocarditis after the vaccination, unsure relation of myocarditis with COVID (e.g., positive serological results of different viruses), and unfulfilling ESC criteria.

### 2.3. Data Extraction and Studies Quality Assessment

Similarly to the study selection, data were extracted independently by two researchers (GI and MJ) and discrepancies were solved by the third (SU). Data regarding demographics and medical history, COVID presentation, applied diagnostics, treatment methods and outcome were collected (Table 1 and Table 2). We have analyzed exclusively case-report studies, which are per se associated with a high risk of bias, given that we have not applied a standardized tool for the quality assessment.

## 3. Results

After the full-text assessment, 59 studies and 70 patients were included in the analysis. All the counted percentages were calculated from the number of patients who had reported specific parameters.

### 3.1. Demographics

Men constituted 50 out of 69 (73%) cases which reported sex. Age was reported in every case; the mean age was 44 (SD 15.67). Only in 1 reported case patient was vaccinated; however, the vaccination scheme was not fulfilled—the patient received one dose of an unspecified vaccine 9 days before admission. Prior medical history and comorbidity status were reported in 46 cases—15 patients (33%) had no previous significant medical history. The most frequent comorbidities were hypertension (*n* = 16, 35%), obesity (*n* = 5, 11%), dyslipidaemia (*n* = 4, 9%) and asthma (*n* = 3, 7%).

### 3.2. Clinical Characteristics

Symptoms were reported in 68 patients. Most frequent symptoms were dyspnoea (*n* = 36, 53%), cough (*n* = 22, 32%), chest pain/discomfort (*n* = 19, 28%), diarrhoea (*n* = 11, 16%) and weakness/fatigue (*n* = 10, 15%). 

Fever status was reported in 43 cases, and the exact value of temperature was reported in 39 cases, mean reported temperature was 38 C (SD 1.42) and 18 patients (42%) were febrile (over 38 C). Heart rate (HR) was reported in 46 cases, and the exact value was shown in 44 cases. The mean HR value was 112 beats per minute (SD 26.5), and tachycardia (over 100) was reported in 33 patients (75%). Blood pressure was described in 44 patients, and the exact value was shown in 40 cases. Mean systolic blood pressure was 105 mmHg (SD 26.17), and diastolic was 66 mmHg (SD 16.83). Hypotension (mean arterial pressure below 65 mmHg) occurred in 15 patients (34%). Oxygen saturation without oxygen supply was reported in 37 cases; the mean value was 90% (SD 8.06), and 21 (57%) patients were considered to have low saturation (below 95%).

### 3.3. Time of COVID-Related Myocarditis

The time relationship between COVID and myocarditis, i.e., myocarditis concurrent with the infection or after a specific period, was reported in 69 cases. In 46 patients (67%), myocarditis occurred during the infection. In the remaining 23 patients (33%), myocarditis followed infection. The time between infection and myocarditis was reported in 20 cases (87% of cases of myocarditis after COVID), and the mean time was 52 days (SD 49.13). 

### 3.4. Laboratory Markers

Method of confirmation of COVID infection was declared in 56 patients. Polymerase chain reaction (PCR) showed the virus presence in 45 cases (80%), COVID was diagnosed based on serology results in 5 patients (9%), biopsy analysis revealed infection in 4 patients (7%), antigen test was performed in 2 cases (4%).

The status of inflammatory markers was reported in 62 patients The reported markers included: C-reactive protein (CRP) (reported in 58 cases, elevated in 54 (93%)); (white blood count (43 cases, elevated in 25 cases (58%)); procalcitonin (20 cases, elevated in 19 cases (95%)); ferritin (18 cases, elevated in 15 (83%)); interleukin-6 (13 cases, elevated in 100%); erythrocyte sedimentation rate (4 cases, elevated in 100%) and fibrinogen (reported and elevated in 1 case).

Most frequently reported cardiac biomarkers included: troponin (reported in 69 cases, elevated in 66 (96%)), brain natriuretic peptide (BNP) or N-terminal BNP (NT-proBNP) (reported in 49 cases, elevated in 47 (96%)) and DDimer (reported in 32 cases, elevated in 29 (91%)).

All the values are from the earliest assessment during patients a hospital stay. We decided not to provide mean or median values of the reported markers due to different normal range values and laboratory assessment methods.

### 3.5. ECG

Electrocardiography results were reported in 58 cases. No pathological changes were reported in 3 patients (5%). Most frequent changes were: ST wave abnormalities (*n* = 22, 38%), sinus tachycardia (*n* = 22, 38%), T wave abnormalities (*n* = 12, 21%), low QRS voltage or low QRS progression (*n* = 10, 17%), ectopic rhythm origin—including atrial fibrillation (*n* = 7, 12%). Ventricular tachycardia occurred in 4 patients (7%).

### 3.6. Imaging Diagnostics

#### 3.6.1. X-ray

Chest X-ray findings have been reported in 30 cases (43%). The radiograph description of the heart has been mentioned in 3 cases, out of which 3 demonstrated cardiomegaly as a cardiac abnormality. Changes in lungs, however, have been reported in 19 case reports (27%), primarily multifocal opacities suggestive of viral pneumonia.

#### 3.6.2. CA

The result of coronary angiography (CA) has been reported in 23 cases (33%). None of them revealed any relevant coronary artery stenosis.

#### 3.6.3. CT

Computed Tomography (CT) has been performed in 42 cases (60%). Changes in heart have been stated in 11 patients (26%), out of which pericardial effusion was described in 6 cases and cardiomegaly in 4 patients. The changes in lungs have been observed in 31 cases (74%), mostly bilateral ground-glass opacities and pleural effusion characteristic of viral pneumonia. 1 case report mentioned evidence of pulmonary embolism in CT findings.

#### 3.6.4. MRI

The findings observed in Magnetic Resonance Imaging (MRI) have been reported in 33 cases (46%). The number of cases fulfilling Lake Louis criteria constituted 22 cases (67%), whilst 11 cases (33%) did not meet Lake Luis criteria. Nonischemic myocardial injury has been observed in 25 cases (76%). Myocardial oedema was confirmed in 17 cases (52%), systolic dysfunction in 11 cases (33%), hypokinesia in 7 cases (21%) and pericardial effusion in 5 of them (15%).

#### 3.6.5. ECHO

The echocardiography results have been mentioned in 66 out of 70 cases (94%). Systolic dysfunction has been described in 55 patients (83%), among which 13 (24%) cases reported on biventricular. The Left ventricular ejection fraction (LVEF) has been measured in 46 cases, and the average LVEF turned out to be 28%. Heart wall hypokinesia or akinesia was reported in 24 cases (36%). The observation of pericardial effusion has been described in 19 cases (29%), heart hypertrophy in 13 cases (20%) and mitral or tricuspidal regurgitations in 6 cases (9%).

### 3.7. Biopsy

Myocardial biopsy has been performed in 23 out of 70 cases (33%). The Dallas Criteria for myocarditis have been fulfilled in 21 (91%). The most commonly reported changes were diffuse lymphocytes infiltration which was found in 21 cases (91%), myocyte necrosis, mentioned in 5 cases (21%) and oedema in 4 described cases (17%).

### 3.8. ESC Criteria

The sufficient information for establishing fulfilment of criteria of clinically suspected myocarditis, according to the European Society of Cardiology (ESC) guidelines [6], has been provided in 68 out of 70 cases (97%). The cases fulfilling only one criterium constituted 3 out of 68 cases (4%). The fulfilment of two criteria has been observed in 29 out of 68 cases (43%), three of them in 25 out of 68 cases (37%) and four of them were met in 11 out of 68 cases (16%). The mean number of fulfilled criteria was 3.

The first criterium, I. ECG/Holter/stress test features was fulfilled in 34 out of 68 cases (50%), the second criterium, II. Myocardiocytolysis markers was fulfilled in 64 out of 68 cases (94%), the third criterium, III. Functional and structural abnormalities on imaging (echo/angiography/CMR) was fulfilled in 58 out of 68 cases (85%) and the fourth one, IV. Tissue characterization by CMR was met in 24 out of 68 cases 35%). 

### 3.9. Treatment

#### 3.9.1. Pharmacotherapy

The analysis of treatment implemented in the care of described patients revealed the applied pharmacotherapy in 65 out of 70 cases (92%). Antibiotics were used in 43 out 65 cases (66%), among which azithromycin in 14, piperacillin/tazobactam in 10, meropenem in 8, vancomycin in 7, doxycycline in 7, colchicine in 7 and ceftriaxone in 6 cases. Steroids have been used in 42 cases (65%), with methylprednisone being most often prescribed—in 24 cases. Vasopressors were used in 24 out of 65 cases (37%). Anti-viral drugs were applied in 18 out of 65 cases (28%), among which remdesivir in 9 and Lopinavir + Ritonavir in 5 cases. Inotropics were used in 16 out of 65 cases (25%), among which dobutamine was prescribed in 10 cases. Furtherly, anti-hypertensive drugs in 14 cases, hydroxychloroquine in 14 cases, IVIG in 14 cases, diuretics in 11 cases, anticoagulants in 11 cases, antiplatelet therapy in 10 cases, tocilizumab in 5 cases, NSAIDs in 4 cases and PPIs in 4 cases. 

#### 3.9.2. Procedures

The information about medical procedures performed in the care of described patients was mentioned in 34 out of 70 cases (49%). Mechanical ventilation has been applied in 14 patients (41%), and breathing support, meaning CPAP, high-flow or just oxygen, in 8 cases (24%). Extracorporeal membrane oxygenation (ECMO) was used in 16 patients (47%), and Continuous Renal Replacement Therapy (CRRT) in 7 patients (21%). Cardiac devices (ex. Impella CP) have been implemented in 10 cases (29%). The performance of pericardiocentesis or pericardiotomy has been mentioned in 6 cases (18%). Resuscitation was performed in 3 out of 34 cases (9%) and heart transplant in 1 out of 34 patients (3%).

### 3.10. Complications

Adverse events complicating the course of illness have been described in 38 out of 70 cases (54%). The cardiogenic shock occurred in 15 out of 38 cases (40%), acute respiratory distress syndrome (ARDS) in 12 (32%), heart failure in 5 (13%), acute kidney injury (AKI) in 5 (13%), mixed distributive and cardiogenic shock in 4 (11%), septic shock in 3 (8%). Other complications mentioned included: disseminated intravascular coagulation (DIC), lower limb ischemia, lower limb myositis, facial nerve palsy and pulmonary infection.

### 3.11. Outcome

The final outcome has been reported in 63 out of 70 cases (90%). In 53 cases (84%), recovery has been achieved. The number of days till the patient’s discharge was stated in 22 out of 53 cases (42%), with the average number of 16 days and the range from 3 to 52 days. Death was the outcome in 11 out of 63 cases (17%). The number of days from admission to the hospital till death has been given in 7 out of 11 cases (64%), with the average number of 9 days with the range from <1 day to 33 days. Most important features are shown in Figure 2. Clinical features by outcome groups are summarized in the harvest plot (Figure 3).

A summary of the most important findings is shown in Table 1 and Table 2 and displayed in Figure 1. The full report of reviewed cases is in Appendix A.

## 4. Discussion

### 4.1. Clinical Characteristics of Patients

Reported patients were predominantly men (73%). Myocarditis occurred in relatively young patients (mean age 44 years). Notably, 33% of patients had no previous medical history. Given that, the phenotype of the myocarditis patient—a mid-aged man with no or a few comorbidities, does not suggest severe illness and can be falsely comforting for a physician. In 33% of cases, myocarditis occurred sometime after the COVID infection. Center for Disease Control and Prevention reports that myocarditis can be diagnosed several months after the infection. In the observed cohort of 36,005,294 patients, 89.6% were diagnosed with myocarditis in the same month as COVID infection, 6.6% 1 month after, and 3.9% ≥ 2 months after [69]. In our case, the mean time between infection and myocardium affection was 52 days. However, one case occurred after 6 months after the initial disease. This highlights the necessity of watchful surveillance of susceptible patients after the COVID, even up to half a year. Most of the published case reports describe patients who experienced myocarditis when the vaccines were not yet available. Only one patient was vaccinated, however, he did not finish the complete vaccination scheme. Further, some studies showed an increased incidence of myocarditis after COVID vaccines [70,71]. A recent analysis performed by the Center for Disease Control and Prevention, which included 40 health care systems, revealed that the risk of myocarditis after vaccination is lower than the risk of myocarditis after COVID infection regardless of sex and age group [72]. The impact of the vaccination on the incidence and severity of myocarditis after COVID infection—despite prior immunization—needs further cohort studies. 

### 4.2. Clinical Presentation

Some of the most reported symptoms were dyspnoea, cough, chest pain and reported weakness. These are the typical myocarditis manifestations, and as they overlap the symptomatology of the COVID infection, making an appropriate diagnosis can be challenging [73]. Noteworthy, we observed a relatively high occurrence of diarrhoea (16%). In every, except one, cases diarrhoea accompanied myocarditis at the time of COVID. Given that, it can be associated rather with the infection than the myocarditis [74].

Inflammatory markers, including white blood count, CRP, procalcitonin, ferritin, and interleukin-6, were typically elevated. The precise pathophysiology of COVID-related myocarditis remains unsolved, and a cytokine storm remains one of the hypotheses [75]. Significant elevation of inflammatory markers (CRP, ferritin, DDimer) were also associated with a more aggressive course [8,76].

Noteworthy, cytokine storm as a reflection of the hyperactive immune response is not the only proposed hypothesis for explaining myocardial injury by COVID-19. COVID molecules invade i.a. heart cells via the protein receptor angiotensin-converting enzyme 2 (ACE2). ACE2 is also associated with the modulation of the myocardial inflammatory response. The affected heart cells were noted to have a higher expression of the ACE2 receptors [77,78]. Given that, the role of the ACE2 receptors in COVID-related myocarditis pathophysiology seems to be significant.

Cardiac markers, such as troponin or NT-proBNP, were elevated in most cases. The rise of both biomarkers in COVID infection was associated with a poor prognosis. Notably, the prognosis of the patients with preceding cardiovascular disease remained favorable [79]. The high troponin level was also associated with the arrhythmia prevelance [80]. Cardiac biomarkers, especially troponin, should be analyzed in the context of other laboratory results. While the isolated rise of troponin reflects the isolated cardiac involvement, the rise of troponin with the surrounding intensive rise of inflammatory biomarkers reflects the hyperinflammatory state with possible multiorgan dysfunciton [8]. This distinction has important clinical implications, as physicians should always recognize which therapeutical pathway would be more rewarding, i.e., targeting inflammation by using steroids, immunoglobulins, biological treatment (e.g., anakinra, tocilizumab, sarilumab, canakinumab, JAK inhibition) or focusing on the cardioprotective treatment [81,82,83,84,85].

Described electrocardiographic findings present a wide range of changes and are not characteristic of myocarditis. On the other hand, physiological ECG was reported only in 5% of cases, suggesting that the ECG can successfully serve as a screening tool for myocarditis.

### 4.3. Imaging Diagnostics

Although endomyocardial biopsy (EMB) has lost its popularity [86], it remains the gold standard in diagnosing myocarditis. As it is an invasive procedure, it requires professional training and has complications, but when done correctly, it has invaluable diagnostic and prognostic value [87,88]. The EMB was performed in 33% of all reviewed cases. Lymphocytic infiltrates associated with myocyte necrosis, which stands for Dallas Criteria, were present in 91% of cases (n = 21). With the evolution of immunohistochemistry technics, more detailed analysis allows detecting more myocarditis cases and overcoming the limitations present in the traditional sample examination [89].

The deterioration of left ventricular systolic function was present in 83% of the described patients. This finding remains consistent with other systematic reviews: Jaisiwal et al.—74% [73], Ho et al.—66% [9]. Similarly to what has been shown in previous systematic review performed by Castiello T. et al. [8], one of the most common echocardiographic findings was global and regional wall hypokinesia 36.4% (*n* = 24), suggestive of myocardial oedema. Even though there are no echocardiographic changes unique to myocarditis, the assessment of echocardiographic parameters is recommended prior to the EMB procedure. It allows excluding pericardial effusion or intracavitary thrombus, as well as it could help to exclude other causes of heart failure and may have a prognostic value [89]. Moreover, the initial ECHO diagnostic is also an excellent parameter to be controlled in follow-up [90].

Cardiac magnetic resonance is considered a noninvasive gold standard for diagnosing myocarditis [91]. Late gadolinum enhancement, indicating the most inflamed areas, was present in 66,7% (*n* = 22) of patients. It is also a prognostic marker for the increased risk of adverse cardiac outcomes in patients with myocarditis [92]. Myocardial oedema, which was present in 51% (*n* = 17) of reviewed cases, may affect myocardial function. It can be an expression of diffuse inflammation due to systematic response, vascular leakage induced by endothelial barrier dysfunction or direct myocardial damage caused by SARS-CoV-2 [93,94].

The most often quoted abnormal finding in X-Ray was cardiomegaly (25%, *n* = 9) which can be a non-specific manifestation of various primary or acquired cardiomiopathies [95]. The most common X-Ray findings were opacities suggesting COVID-19 pneumonia (61% *n* = 19). Normal X-Ray findings were described in 12% (*n* = 4). The 22 patients (31%) who underwent CA were predominantly men (*n* = 16), and their age ranged from 19 to 69, with a median of 42.5 years. Although CA is not a standard procedure in the diagnostics of myocarditis, the patients presenting acute retrosternal pain and dyspnoea at a relatively young age raise the alarm for myocardial infarct procedures [96].

### 4.4. Treatment

The pharmacotherapy strategies varied among the described patients, depending on comorbidities and centre-specific procedures. It is impossible to enucleate specific myocarditis treatment, as the strategy should focus first on the disease source and unique patient’s symptomatic [89].

One of the main targets for the pharmacotherapy of COVID-19 myocarditis was cytokine storm. Incorporating the steroids (65% of our cases) into the COVID treatment was an attempt to address this pathological pathway. Indeed, the RECOVERY platform trial has proven that steroids play an important role in reducing COVID-19 mortality [97]. Dexamethasone was also found to diminish myocardial oedema and improve systolic function [93]. Another drug considered to dump patients’ immune response was tocilizumab (5 reviewed cases), which was shown to reduce mortality even in the group already treated with steroids [84,85,98]. Intravenous immunoglobulins were used in 14 patients, yet their use to modulate inflammation in COVID is still under debate [99,100]. Other conceptions for the immunomodulatory treatment in COVID tested interleukin-1β inhibitor—canakinumab; and colchicine. These conceptions did not show clinical or prognostic benefit in COVID setting [83,101]. 

Treatment with non-steroidal anti-inflammatory drugs (NSAIDs) in myopericarditis is another unsolved problem. There are studies supporting the hypothesis that it can be potentially harmful and should be avoided in myocarditis [102], as well as studies where NSAIDs neither affected all-cause mortality rate nor left ventricle function [103]. Nevertheless, none of the studies showed the benefit of NSAIDs in myocarditis treatment, and generally, they should not be widely implemented [6]. 

As ARDS was the most common complication, antibiotics were used, i.a., to prevent bacterial superinfection. The most willingly used antibiotic was azithromycin [9,73,104]. It seems that using antibiotics in standard COVID care is unjustified [105]. The anti-viral treatment was applied in 27% of the reviewed cases. None of the tested antiviral agents has proven its efficacy in improving COVID survival [106,107,108]. Further, there is a lack of evidence about antiviral treatment for myocarditis in general [89,109]. Thus, the symptomatic treatment according to NYHA functional class (with beta-blockers, diuretics, angiotensin-converting enzyme inhibitors or angiotensin-II receptor blockers) should be applied [6,89].

Surgical procedures like mechanical circulatory support or heart transplant are warranted particularly for patients with cardiogenic shock in the course of fulminant myocarditis, whose clinical state is deteriorating despite optimal medical treatment. Similarly, cardiac devices implementation should be considered in patients with myocarditis-related atrioventricular blocks or after a myocardial infarct. These technics should be considered early, when the highest possible pharmaceutics doses are insufficient to facilitate recovery [6,89].

### 4.5. Prognosis and Outcome

The overall incidence of combined in-hospital mortality and heart transplantations associated with myocarditis, presented in 2018 in the study from Multicenter Lombardy Registry, was 3.2% [110], much lower than in our work (17% of deaths). Higher mortality was also reported in the corresponding systematic reviews [8,9,73,104], ranging from 15.2% to 31.8%. It can be explained by the specific course of COVID-related myocarditis, which is overlapped by the aggressive, infectious disease. Noteworthy, case reports are the low-quality evidence and reported cases are usually severe and complicated, which may artificially increase the reported mortality. Given that, forthcoming registries are warranted to establish reliable COVID-related myocarditis mortality.

### 4.6. Limitations

Our study is not free from limitations. The most important one is that the data for our analysis come from the case-report papers, which provide poor quality evidence. Further, most of the reported cases demonstrate very dramatic situations—patients with severe and sophisticated course and complications—which is attractive for the publication reasons but does not reflect the typical clinical situation. Given that, the generalizability of our conclusions is limited, especially in epidemiology. Conversely, presented descriptions give a detailed, patient-level evaluation of each case. Submitted data was frequently incomplete—not every parameter was regularly described. We tried to choose and analyze only the most repetitive factors. Laboratory parameters were unsuitable for statistical analysis, as laboratories use different evaluation techniques and normal range values. Considering it all, quality or bias assessments could not be performed. Our study provides a current, interim summary of the data, while the results of more extensive, well-designed studies are unavailable.

## 5. Conclusions

This systematic review provides the most recent summary of the reported COVID-related myocarditis. We present the complex summary of COVID-related myocarditis, which can serve as a reliable source of knowledge on that topic. Importantly, in approximately one-third of patients, myocarditis occurs a few weeks after infection, even up to 6 months. The necessity of considering COVID as a cause of myocarditis in relatively young and healthy populations should be strongly highlighted. Susceptible patients should be carefully observed after the COVID infection, even the poorly symptomatic one.

## Figures and Tables

**Figure 1 jcm-11-05519-f001:**
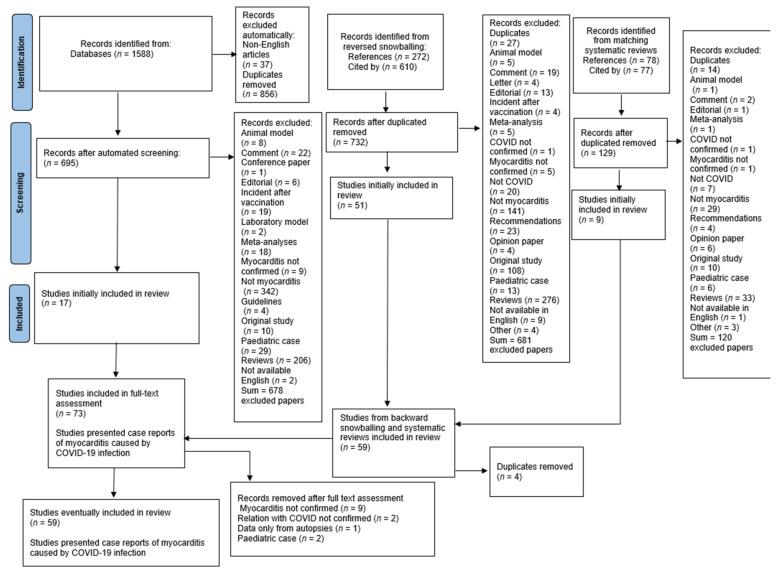
Flow chart of the systematic literature search according to PRISMA guidelines.

**Figure 2 jcm-11-05519-f002:**
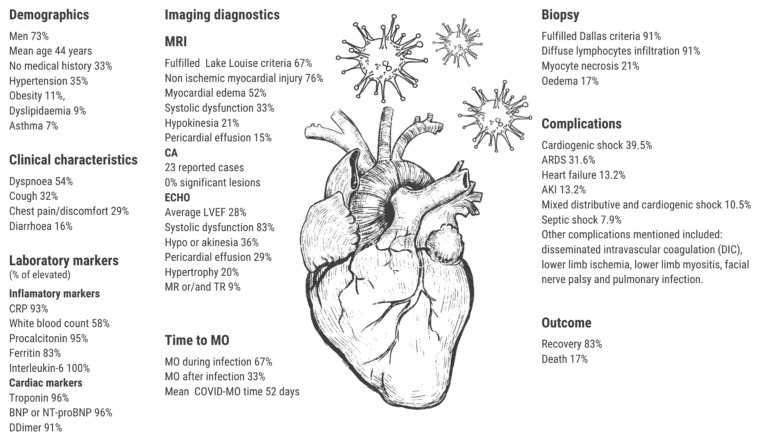
The central illustration of the findings. Abbreviations: MO—myocarditis, CRP—c-reactive protein, NT-proBNP—N-terminal brain natriuretic peptide, ARDS—acute respiratory distress syndrome, AKI—acute kidney injury, MRI—magnetic resonance imaging., CA—coronary angiography, ECHO—chocardiography.

**Figure 3 jcm-11-05519-f003:**
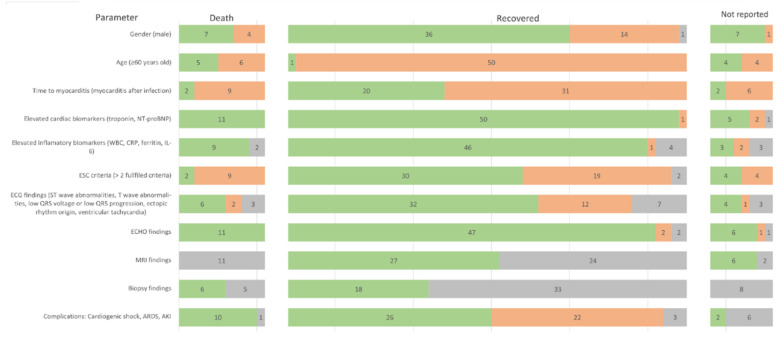
Harvest plot summarising the most important features of the studied patients by groups of outcomes. Green colour indicates the presence of the feature, orange its absence, and grey a missing value. The numbers on the bars show the number of patients in each group. Abbreviations: NT-proBNP—N-terminal brain natriuretic peptide, WBC—white blood count, CRP—c-reactive protein, IL-6—interleukin 6, ESC—European Society of Cardiology, ECG -electrocardiogram, ECHO—echocardiography, MRI- magnetic resonance imaging, ARDS—acute respiratory distress syndrome, AKI—acute kidney injury.

**Table 1 jcm-11-05519-t001:** Demographics, clinical characteristics and laboratory findings of studied patients. All descriptive parameters are obtained from the original papers.

Author (One Surname et al.)	Age (Years)	Sex (m/f)	Reported Symptoms	Medical History (Comorbidities, Previous Surgeries, Events)	Myocarditis during COVID (yes/no)	Myocarditis Post COVID (How Long)	Troponin	NT-probnp	ECHO Findings
Doyen et al. [10]	69	m	vomiting and diarrhoea 7 days before, cough, fever of 39 °C, dyspnoea,	hypertension	yes	no	9002 ng/L (<40)	NR	mild LVH
Luetkens et al. [11]	79	m	fatigue, shortness of breath, recurrent syncopes,	asthma	yes	no	18.8 ng/L ->63.5 ng/L	normal → 1178 pg/mL	normal
In-Cheol et al. [12]	21	f	febrile sensation, coughing, sputum, diarrhoea, shortness of breath	NR	yes	no	1.26 ng/mL (<0.3)	1929 pg/mL (<125 pg/mL)	severe LV systolic dysfunction
Sala et al. [13]	43	f	oppressive chest pain, dyspnoea	NR	yes	no	135-107-106 ng/L (<14)	512 pg/mL (<153)	mild LV systolic dysfunction, LVEF 43%, with inferolateral wall hypokinesis
Bonet et al. [14]	19	m	recurrent fever, cough, cervical adenopathy, chest pain and dyspnoea	NR	no	4 weeks	4200 ng/L (<79)	17,377 (<500)	LVEF 20%, low cardiac output
Ishikura et al. [15]	45	m	NR	NR	no	5 weeks	31.516 pg/mL (≤30)	NR	severe hypokinesis, LVEF 7.4%
Meyer et al. [16]	81	m	no symptoms	hypertension, prostate adenocarcinoma	no	6 months	NR	NR	LVEF 39%, global mild hypokinesia and severe inferior wall hypokinesia
Roest et al. [17]	50	NR	acute symptoms of cardiac decompensation	6 years post heart transplantation	no	5,5 months	55 ng/L	212 pmol/L (elevated)	biventricular failure and congestion
Inciardi et al. [18]	53	f	severe fatigue, fever and cough week before	No previous history of cardiovascular disease	yes	no	0.24 ng/mL (<0.01)	5647 pg/mL (<300)	LVEF 40%, increased wall thickness, the diffuse echo-bright appearance of the myocardium, diffuse hypokinesis, LV diastolic function mildly impaired with mitral inflow patterns, E/A ratio of 0.7 and E/e’ ratio of 12, pericardial effusion (max 11 mm) without signs of tamponade
Hu et al. [19]	37	m	chest pain and dyspnoea for 3 days, accompanied by diarrhoea	NR	yes	no	>10,000 ng/L (elevated)	21,025 ng/L (elevated)	LVEF 27%, enlarged heart LVED 58 mm, LAED 39 mm, RVED 25 mm, RAED 48 mm, trace (2 mm) pericardial effusion
Zeng et al. [20]	63	m	expectoration of the white sputum, shortness of breath, chest tightness after activity,	allergic cough for 5 years, previous smoking history	yes	no	11.37 g/L (elevated)	22.600 pg/mL (elevated)	LVEF 32%, diffuse myocardial dyskinesia, enlarged left ventricle, pulmonary hypertension 44 mmHg
Aldeghaither et al. [21]	39	f	fever, dyspnoea, chest pain, diarrhoea	NR	No	4 weeks (asymptomatic)	11,810 ng/L	NR	large pericardial effusion with severe biventricular dysfunction and LVEF of 10–15% (cardiac tamponade with concurrent cardiogenic shock)
Aldeghaither et al.	25	m	dyspnoea, fever, hypotension	NR	no	5 weeks (asymptomatic)	1557 ng/L	NR	severe LV systolic dysfunction with LVEF 15–20% and moderate RV systolic dysfunction
Aldeghaither et al.	21	m	dyspnoea, fever	NR	yes	no	27,000 ng/L	NR	severe biventricular systolic dysfunction with LVEF 5–10%
Amin et al. [22]	18	f	loss of consciousness, fever, headache	NR	yes	no	3 × UNL	21,000 pg/mL (<125)	small LV, LVEDD: 42 mm, LVEF: 10%with global hypokinesis, a normal RV, and PAP, no PE or thrombosis, and no significant valve lesions or dysfunction
Amin et al.	32	m	syncope after an episode of chest pain with fever and myalgia, dyspnoea, dry cough	NR	yes	no	2 × UNL	10,000 pg/mL (<125)	LVEF: 25%, an enlarged LVEDD, normal RV size (mild dysfunction), and normal valvular and pericardial function, increased wall thickness.
Amin et al.	34	f	losing consciousness, fever and headache for six consecutive days	NR	yes	no	4 × UNL	3000 pg/mL (<125)	normal LV and RV size, moderate TVR and MVR, sPAP: 30 mmHg, LVEF: 30%.
Amin et al.	22	m	dry cough, dyspnoea	no previous medical history	yes	no	3 × UNL	21,000 pg/mL (<125)	LVEF 30–35%, LVEDD of 5 cm, moderate RV enlargement and dysfunction, dilated inferior vena cava, and mild left-sided pleural effusion
Amin et al.	45	m	fever, dyspnoea, pleuritic chest pain	NR	yes	no	2 × UNL	4300 pg/mL (<125)	LVEF of 45%, normal LV size and function without regional wall motion abnormalities, and circumferential pericardial effusion without significant respiratory changes in the mitral and tricuspid valves
Amin et al.	28	m	chest pain, fever, dry cough	NR	yes	no	2 × UNL	1200 pg/mL (<125)	LVEF 30–35%, hypokinesis in the anterior wall, normal LV size as well as normal RV size and function
Amin et al.	51	f	fever, cough, cardiopulmonary arrest	NR	yes	no	3 × UNL	22,000 pg/mL (<125)	LVEF 35%, normal LV size, increased LV wall thickness
Amin et al.	39	f	chest pain, cough, orthopnea	asthma	yes	no	2 × UNL	23,000 pg/mL <125 pg/ml	LVEF 30–35%, mild pericardial effusion, RV enlargement, hypermobile thrombus in the LV (1.5 × 1.5 cm)
Menter et al. [23]	47	f	oligosymptomatic flu-like disease for a week, then—she was found unconscious and apneic at home	obesity [body mass index (BMI) 31.6], depression	yes	no	507 ng/L (elevated)	>70,000 ng/L (elevated)	moderately reduced left ventricular function LVEF 30% and normal right ventricular size and function. There was no evidence of left ventricular hypertrophy or dilatation.
Ashok et al. [24]	53	m	fever and right upper quadrant abdominal pain	no previous medical history	yes	no	2.6 ng/L (normal)	NR	LV systolic impairment with no obvious regional wall motion abnormalities
Li et al. [25]	60	m	8 days of fever, cough, and worsening dyspnoea, with mild abdominal pain and diarrhoea despite completing a 3-day course of azithromycin as an outpatient. Denied chest pain, nausea, or vomiting.	hypertension and hyperlipidaemia	yes	no	582 ng/L (<6–14)	15,642 pg/mL (0–300)	significant for severe segmental LV systolic dysfunction, LVEF 15–20% with hypokinesis of the apex, distal anterior septum, and anterior and lateral walls, with a small pericardial effusion. Right ventricular size and function were normal
Nakatani et al. [26]	49	m	NR	no previous medical history	yes	no	1.010 ng/mL (≤0.014)	27,541 pg/mL, (≤125.0)	low LVEF: <20% and diffuse myocardial hypertrophy
Ismayl et al. [27]	53	m	chest pain, diaphoresis, intense fatigue, and body aches	no previous medical history	no	35 days	2.83 ng/mL (<0.04), with a peak of 13.6 ng/mL	60,230 pg/mL	severe global hypokinesis of the left ventricle LVEF: 25%
Thomson et al. [28]	39	f	1-day history of profuse diarrhoea, vomiting, abdominal pain, and no respiratory symptoms	laparotomy with ovarian cystectomy	yes	no	118 ng/L (<14)	6543 (norm: <125)	severely reduced biventricular function, normal cardiac chamber size, and a globally thickened, bright myocardium (measuring 16 mm in the septum and inferolateral wall) with a small-to-moderate-sized pericardial effusion.
Campoamor et al. [29]	32	m	acute chest pain, palpitations	After COVID-19, only dry cough and odynophagia were exhibited, resolved after symptomatic treatment with paracetamol. otherwise healthy	no	40 days	149.87 pg/mL (24–30 pg/mL)	70.89 pg/mL (<125)	a dilated LV, mild diffuse hypokinesia with light impairment of global systolic function, and a dilated LA
Flagiello et al. [30]	30	f	fever, chest pain, and worsening dyspnoea for 5 days	No previous medical history	yes	no	5891 ng/L (elevated)	16,569 ng/L (elevated)	20-mm pericardial effusion compressing the right heart chambers and LV systolic dysfunction, LVEF: 35%, with global hypokinesis
Pascariello et al. [31]	19	m	worsening of the flu syndrome, presented with fever, cough and diarrhoea and vomits for 3 days.	autism spectrum	yes	no	1033 ng/mL (<14)	47,650 ng/mL (<300)	enlarged LV: 56 mm, diffuse myocardial hypo-akinesia along with severe left ventricular ejection systolic dysfunction LVEF: 15–20%, decrease in right cardiac function
Gaudriot et al. [32]	38	m	chest pain and vomiting	suggested chronic lymphopenia	no	5 weeks	1600 pg/mL (<14)	10,500 pg/mL (<300)	biventricular infiltrative myocardial hypertrophy, associated with a circumferential non-compressive pericardial effusion and a restrictive mitral inflow profile, no systolic dysfunction
Shah et al. [33]	19/38	m/m	7-day history of fever, generalized weakness, cough, and shortness of breath/chest pain, not related to breathing, exertion, or posture	no previous medical history	yes/no	no/3 weeks	1. 256 ng/mL (<15)/2. 1264 ng/L (<15)	NR	normal left ventricle size with severe global hypokinesia and severely reduced LV systolic function LVEF:24%, and there was mild-to-moderate MR, RV function was also severely reduced, with moderate-to-severe TR and mildly increased PAP 40 mm Hg, no pericardial effusion/normal left ventricular systolic function, LVEF: 53%, apex was noted to be akinetic while other segments were moving normally, no pericardial effusion
Shabbir et al. [34]	50	f	4-day history of central chest pain, which was made worse on lying flat and on deep inspiration	hypertension, previous myocarditis, reactive arthritis	no	no	77 ng/L (0–14)	NR	a trivial anterior pericardial effusion with good biventricular function
Gauchotte et al. [35]	69	m	fever, asthenia, abdominal pain progressing for a week.	diabetes, hypertension, ischemic heart disease	no	no	8066 pg/mL (elevated)	NR	a non-dilated LV and severe and diffuse LV hypokinesia, LVEF: 30%
Bernal-Torres et al. [36]	38	f	palpitations that started suddenly 3 days prior to presentation and were associated with haemodynamic instability	no previous medical history	yes	no	1190 ng/mL (<14)	BNP 13,000 pg/mL (<100)	LV with global hypokinesia, severely reduced systolic function, LVEF:30%, without valvular heart disease, and mild pericardial effusion (2 mm)
Oleszak et al. [37]	52	m	cough, subjective fever, shortness of breath, trace hemoptysis over the last 5 days	hypertension, previous myocarditis, reactive arthritis	yes	no	0.017 ng/mL (elevated)	proBNP 1220 pg/mL (elevated)	LVEF: 10–15%, reduced RV systolic function, and global dilatation of all 4 chambers, RV, RA and LA moderately dilated, compared to mild LV dilation, LVIDd: 5.8 cm, lLVIDs: 5.1 cm, IVS: 1.3 cm, sPAP: 18 mm Hg, no significant valvular pathology, mild MVR and TVR, pericardial effusion was not visualized
Gay et al. [38,39]	56	m	dyspnoea and lethargy	obesity, hyperlipidemia	yes	no	troponin I: 1.3 ng/mL (elevated)	BNP 790 pg/mL (elevated)	extreme concentric LVH with a wall thickness of 2.0 cm, reduced biventricular function, LVEF: 20%, and a small pericardial effusion
Gomila-Grange et al. [39]	39	m	fever, right flank pain and diarrhoea for 6 days	former smoker, 68 g daily alcohol use	yes	no	troponin I: 3800 ng/mL (<13)	pro-BNP 27,696 pg/mL (< 125)	biventricular dysfunction, LVEF: 30% due to diffuse hypokinesis, diastolic dysfunction and slight posterior pericardial effusion without tamponade signs
Richard et al. [40]	28	f	found lethargic and covered in coffee ground emesis at home	diabetes mellitus type 1, diabetic gastroparesis, asthma, anxiety, depression with multiple previous episodes of DKA, questionable history of IV drug use	no	yes (time not reported)	0.04 ng/mL (<0.04)	NR	LVEF: 26–30% and MVR
Wenzel et al. [41]	39/36	m/m	shortness of breath	Both patients were obese and had a history of upper airway infection with headache, fever, and cough up to 4 weeks before admission. A; dyslipidemia B: hypertension, smoking dyslipidemia, CAD heart failure, lung disease	no	approx 30 days	0.372/0.057 ng/mL (<0.024)	BNP—109/258 ng/L (<100)	signs of LV dysfunction (decreased global and regional longitudinal strain or reduced LVEF and increased LVEDD) A—preserved LVEF: 60% without any wall motion abnormalities, but the focal echo-bright appearance of the interventricular septum (not shown) and slightly impaired global longitudinal strain.
Bemtgen et al. [42]	18	m	hyperpyrexia (42 °C), chills and tachycardia	NR	no	2 months	341 ng/L, (<14)	NR	severely impaired LVEF, LVEF: 25%
Beşler et al. [43]	20	m	febrile sensation and chest pain	NR	yes		0.572 ng/mL (<0.045)	127 ng/L (<125)	NR
Meel et al. [44]	31	m	exertional dyspnoea	NR	no	3 weeks	319 ng/L (< 14)	143 ng/L (<300)	preserved LV and RV contractility and no regional wall motion abnormality ormyocardial hypertrophy.
Paul et al. [45]	35	m	chest pain and fatigue	NR	yes	no	2885 ng/L (elevated)	NR	normal systolic function with no pericardial effusion
Coyle et al. [46]	57	m	shortness of breath, fevers, cough, myalgias, decreased appetite, nausea, and diarrhoea for 1 week	hypertension	yes	no	0.02 → 7.33 ng/mL (<0.05)	859 pg/mL (<126)	moderate diffuse hypokinesis withrelative apical sparing and an LVEF: 35% to 40%
Irabien-Ortiz et al. [47]	59	f	feverish feeling for 5 days, accompanied by squeezing anginal chest pain, no respiratory symptoms	hypertension, cervical degenerative arthropathy, chronic lumbar radiculopathy, lymph node tuberculosis diagnosed due to erythema nodosum, and migraine	yes	no	220 → 1100 ng/dL (elevated)	4421 ng/L (elevated)	disclosed moderate concentric hypertrophy, diminished intraventricular volumes with preserved LVEF, no segmental abnormalities, and moderate pericardial effusion with no signs of hemodynamic worsening 2 h after admission—severe biventricular failure and diffuse myocardial oedema
Frustaci et al. [48]	50	f	fever, dry cough and shortness of breath	no	yes/no	no	0.077 mcg/L, (<0.014)	NR	normal size and function of the LV with mild pericardial effusion
Sardari et al. [49]	31	m	dyspnoea on exertion and low-grade fever	NR	no	3 weeks	<0.03 ng/mL (normal)	NR	mild LV dysfunction
Warchoł et al. [50]	74	m	hemodynamically unstable new onset VT- lasting 12 h	atrial fibrillation, catheter ablation 3 times, hypertension, type 2 diabetes, and hypothyroidism	yes	no	troponin T 72 ng/L (<14 ng/L)	2451 ng/L (<125)	NR
Shahrami et al. [51]	78	m	shortness of breath, cough, anosmia, and myalgia	hypertension	yes	no	103 ng/mL (0–0.3)	NR	reduced LVEF: 15%, PAP = 50 mmHg, mildly enlarged LV and moderate to severe LV dysfunction, mild diastolic dysfunction, mild MVR, and normal septal thickness
Gaine et al. [52]	58	m	recent-onset palpitations and progressive dyspnoea	no previous medical history	yes	no	T 25 ng/L (0–14)	3428 pg/mL (0–400)	severely impaired LVEF: 20%, MR
Taouihar et al. [53]	51	m	intense epigastric pain at rest and effort, associated with nausea and vomiting progressing for 5 days	hyperthyroidism	yes	no	20,000 ng/L (<20)	NR	akinesia of the tip of the anteroseptal and inferior wall of the LV with severe hypokinesia of the inferior wall, a systolic dysfunction of LV, LVEF: 40%
Dahl et al. [54]	37	m	fever, headache, unilateral, painful neck swelling	no previous medical history	yes	no	T 90 ng/L (0–15)	160 ng/L (0–85)	good ventricular function, no hypokinesia and significant valve pathology at admission. Echo repeated on day 2 revealed a deterioration of the LV function with reduced LVEF: 40%
Matsumura et al. [55]	41	m	general fatigue and a high fever	hypertension	no	no	0.094 ng/mL	4938 pg/mL (elevated)	impaired LV systolic function, LVEF: 53%, LVDDand LVDS: 42 and 31 mm, respectively, IVS: 12 mm; LVPW: 12 mm, and mild pericardial effusion
Praet et al. [56]	31	f	fever, cough, rapidly progressive shortness of breath, an episode of cramping abdominal pain and diarrhoea one week earlier	no previous medical history	no	2 months before	hs-TnT 151 ng/L (elevated)	25,386 pg/mL (elevated)	severely depressed LV function due to moderate to severe hypokinesia
Masiak et al. [57]	43	m	anosmia and ageusia for 1-day, mild fever for a few days. All symptoms disappeared completely. Five weeks later admitted to the hospital due to fever with sweats, sore throat, fatigue, dyspnoea, dry cough, skin changes for 8 days prior to the hospitalization. Additionally, discomfort in the right lower abdomen, discoloured stools, and dark-coloured urine.	no previous medical history	no	5 weeks later	0.119 ng/mL (<0.03)	BNP 965 pg/mL (<7)	globally reduced myocardial contractility, decreased LVEF: 40%
Noori et al. [58]	44	m	dry cough associated with generalized body ache but denied any chest pain, shortness of breath, palpitation, orthopnea, or fever	NSTEMI 2 weeks before, COVID pneumonia 1 month ago	no	1 month	11.67 (elevated)	NR	LVEF: 40–45%, mildly decreased globular LV systolic function and moderate to severe hypokinesis involving inferior, inferoposterior wall without thinning of myocardium
Okor et al. [59]	72	f	worsening shortness of breath,	hypertension, chronic obstructive pulmonary disease	no	1 week	1.0 ng/mL	2336 pg/mL (0–99)	severely reduced LV systolic function, LVEF: 20%, multiple LV wall abnormalities including akinetic inferolateral and apical anterior walls and hypokinetic basal and septal walls, small pericardial effusion was also noted, initial echocardiogram in 4-chamber view showed multiple segmental abnormalities and an LVEF: 20%. Parasternal long axis view showed multiple segmental abnormalities, trace effusion, and low LVEF: 20%.
Bulbul et al. [60]	49	f	cough, shortness of breath	NR	yes	no	9 ng/L (3–10)	430 pg/mL (<125)	significant reduction in the LVEF: 25%, diffusely hypokinetic walls and a reduction in RV function.
Khatri et al. [61]	50	m	fevers, chills, generalized malaise, non-productive cough, dyspnoea for 3–4 days and an episode of near-syncope	hypertension and ischemic stroke.	yes	no	544 ng/L	NR	severe global LV systolic dysfunction, RV enlargement, RV systolic dysfunction, moderate to large pericardial effusion, Intermittent RVimpaired filling and collapse (s/o tamponade)
Nicol et al. [62]	40	m	fever, odynophagia, and left neck pain	obesity (BMI 34.8)	no	4 weeks	485 ng/L (<34)	2960 ng/L (elevated)	decrease in LVEF: 45%, low cardiac output (3 L/min), and both subtle hypertrophy and akinesia of posterolateral left ventricular wall with small pericardial effusion opposite
Hudowenz et al. [63]	48	m	high-grade fever, dyspnoea and haemoptysis	NR	yes	no	3264 pg/mL (0–14)	12,232 pg/mL (1–300)	highly reduced LVEF: 22%, RVEF 28%
Jacobs et al. [64]	48	m	fever, diarrhoea, cough, dysosmia, and dyspnoea	hypertension	yes	no	14,932 ng/L (<45)	9223 pg/mL (<125)	hyperdynamic ventricular function, although under massive support of inotropic agentsand vasopressors
Hussain et al. [65]	51	m	dry cough, fatigue, dyspnoea, and a fever	hypertension	yes	no	0.29 ng/mL (elevated)	1287 pg/mL (elevated)	enlarged heart with a decrease in systolic function and an LVEF: 20%
Rehman et al. [66]	39	m	midsternal chest pain	NR	yes	no	5.97 ng/mL (elevated)	379 pg/mL	LVEF: 55–60%, no wall motion abnormalities, no evidence of pericarditis or pericardial effusion
Monmeneu et al. [67]	43	m	fever for 14 days, dry cough, haemoptoic sputum,	NR	yes	no	T 9 → 24 → 24 → 18 → 15 ng/L (<14 ng/L)	456 pg/mL (<125)	NR
Tavazzi et al. [68]	69	m	worsening dyspnoea, persistent cough, weakness for 4 days,	NR	yes	no	4332 ng/L (elevated)	NR	LVEF: 34%, LVEDD: 56 mm, severe and diffuse LV hypokinesia

**Table 2 jcm-11-05519-t002:** Magnetic resonance, biopsy, complications and outcomes of studied patients. All descriptive parameters are obtained from the original papers.

Author (One Surname et al.)	MR Findings	Biopsy Findings	Complications	Outcome
Doyen D. et al.	subepicardial late gadolinium enhancement of apex and inferolateral wall	NR	NR	NR
Luetkens et al.	mild systolic dysfunction (LVEF: 49%) with discrete global hypokinesis, pericardial effusion localized mainly around the LV lateral wall (~10 mm), diffuse interstitial myocardial oedema with an increased T2 signal intensity ratio, prolonged T1 relaxation times	NR	NR	NR
In-Cheol et al.	diffuse high signal intensity in the LV myocardium, myocardial wall thickening (which suggested myocardial wall oedema), extensive transmural late gadolinium enhancement	NR	NR	NR
Sala et al.	recovery of systolic function (day-7), with the persistence of mild hypokinesia at basal and mid LV segments, diffuse myocardial oedema, determining wall pseudo-hypertrophy,	diffuse T-lymphocytic inflammatory infiltrates (CD3+ >7/mm^2^) with huge interstitial oedema and limited foci of necrosis	NR	recovered
Bonet et al.	NR	myocarditis with inflammatory infiltrates consisting of a majority of lymphocytes and neutrophils, oedema but not typical myocyte necrosis, CD-4-positive T lymphocytes, RT-PCR for SARS-CoV-2 and PCR for other cardiotropic viruses was negative	NR	recovered
Ishikura et al.	NR	diffuse lymphocyte infiltration in the interstitium of the myocardial tissue, atrophy and shedding degeneration myofibrils, no myocardial necrosis	cardiogenic shock	recovered
Meyer et al.	focal infero-basal LV wall oedema, elevated T1 and T2 myocardial relaxation times, the post-contrast sequences showed sub-epicardial and mid-wall late gadolinium enhancement in the basal inferior, basal inferolateral, and anterior LV walls	NR	NR	NR
Roest et al.	LVEF: 22%, late gadolinium enhancement images showed a pattern of extensive subepicardial and several midwall subepicardial spots of enhancement of the left ventricle, myocardial oedema was absent (post-myocarditis state without active myocarditis)	focal subendocardial fibrosis and no signs of clinical relevant rejection	heart failure	recovered
Inciardi et al.	increased wall thickness with diffuse biventricular hypokinesis, especially in the apical segments, severe LV dysfunction (LVEF: 35%), biventricular myocardial interstitial oedema, phase-sensitive inversion recovery sequences showed diffuse late gadolinium enhancement extended to the entire biventricular wall, pericardial effusion was confirmed,	NR	heart failure	recovered
Hu et al.	NR	NR	cardiogenic shock and pulmonary infection	recovered
Zeng et al.	NR	NR	septic shock and DIC	death on the 33rd day of hospitalization
Aldeghaither et al.	NR	eosinophilic infiltrate of the myocardium (infiltrate consisting mainly of mononuclear cells—few lymphocytes and histiocytes with many eosinophils)	cardiogenic tamponade with concurrent cardiogenic shock	discharged home 28 days after admission
Aldeghaither et al.	NR	mixed inflammatory infiltrate with lymphocytes, histiocytes, neutrophils, eosinophils	cardiogenic and vasodilatory shock	discharged home 23 days after admission on intermittent hemodialysis, which was stopped 3 weeks later after improvement of the renal function
Aldeghaither et al.	NR	mild lymph-histiocytic interstitial myocardial infiltrate with occasional neutrophil	cardiogenic and vasodilatory shock	recovery. After 14 days of hospitalization, the patient was discharged home. At this time, both MCS devices were explanted.
Amin et al.	NR	NR	NR	death. she died due to cardiac arrest (less than 24 h after admission)
Amin et al.	NR	NR	NR	recovery. discharged after 2 weeks with an ejection fraction of 45%, Follow-up of the patient for 3 months revealed an LVEF of 45% and normal RV function.
Amin et al.	NR	NR	NR	recovery. Follow-up of the patient for 3 months showed an LVEF of 45–50%
Amin et al.	NR	NR	NR	recovery. discharged after 7 days with stable hemodynamics. Follow-up of the patient for 3 months showed an LVEF of 45% and a normal cTnI value.
Amin et al.	NR	NR	NR	recovery. discharged in a satisfying condition. Follow-up of the patient for 3 months showed an LVEF of 45%, mild pericardial effusion, and normal right ventricular size and function.
Amin et al.	NR	NR	NR	recovered within 3 days and was discharged in a satisfying condition. Follow-up of the patient for 3 months revealed LVEF of 45–50% and a normal cardiac function
Amin et al.	NR	NR	NR	recovery, discharged in a satisfying condition after 10 days. Follow-up of the patient over 3 months demonstrated an LVEF of 50%
Amin et al.	myocardial oedema and hyperemia suggestive of active myocarditis	NR	NR	recovery. discharged in a satisfying condition
Menter et al.	NR	multifocal inflammatory infiltrates consisting of neutrophilic granulocytes, lymphocytes and histiocytes, capillarostasis, and perifocal single-cell necroses of cardiomyocytes		the patient died of cardio-respiratory failure within 48 h of admission
Ashok et al.	NR	no	NR	recovery
Li et al.	NR	NR	-	Recovery. The patient was ultimately discharged for physical rehabilitation on hospital day 52.
Nakatani et al.	NR	Mild lymphocytic infiltration and moderate to severe perivascular fibrosis with wall thickening of intramural arterioles, no sign of severe myocardial injury compatible with typical active myocarditis, ischemic changes were found with a focal coagulative necrotic area at microvascular level (approximately 0.08 mm^2^ in area) with losing nuclei, accompanied by microthrombi with fibrin and platelets in small vessels, scattered megakaryocytes were also seen in the capillaries, microthrombi were seen throughout the specimens and myocytes in non-necrotic areas often showed diffuse cytoplasmic vacuolization, presence of platelets in obstructive and non-obstructive microthrombi within the lumens of microvessels was confirmed by immunohistochemical expression of CD61, fibrin-rich microthrombi were also present as confirmed by phosphotungstic acid-hematoxylin stain, microvessels including intramural arterioles often showed swollen endothelial cells	Cardiogenic shock	death on day 5 due to intractable cardiogenic shock
Ismayl et al.	NR	diffuse interstitial and perivascular neutrophilic and lymphocytic infiltration with rare eosinophils and rare myocyte necrosis, suggesting fulminant myocarditis	combined right and left ventricular failure, multiorgan failure	significant improvement in hemodynamics and ejection fraction over the next 5 days
Thomson et al.	NR	A subtle mild interstitial infiltrate consists primarily of CD68 + macrophages and a lesser number of CD3+ T cells. Microthrombi were identified, with the degradation products suggesting a sub-acute microangiopathic process. No fibrosis nor myocardial necrosis was present. No viral particles were seen on electron microscopy. Concerning other causes of myocarditis, the EMB did not show interstitial fibrosis, iron accumulation, evidence of sarcoid or amyloid, or changes consistent with hypertrophic obstructive cardiomyopathy.	Pulseless electrical activity cardiac arrest with ECMO and CPR. Distal perfusion cannulae placed. Development of ischaemic right lower leg (without ECMO cannulae) requiring DSA and fasciotomy.	death after 9 days of hospitalization
Campoamor et al.	extensive subepicardial fibrosis with diffuse pericardial contrast uptake	NR	none	Recovery. The patient remained asymptomatic for the rest of the hospitalization period and was discharged home after 22 days of hospitalization
Flagiello et al.	area of myocardial late enhancement compatible with the diagnosis of myocarditis	slight interstitial fibrosis without inflammation, Immunohistochemical staining (CD3, CD20, and CD68) confirmed the absence of interstitial inflammation	Cardiogenic shock in less than 12 h with signs of systemic hypoperfusion (systemic arterial pressure <60 mm Hg, sweating, vomiting, oliguria, and lactate 4.5 mmol/L) and further degradation of the left ventricular systolic function (LVEF 10%, left ventricular outflow tract time-velocity integral 6 cm) despite an escalation of the inotropic (dobutamine 10 μg/kg/min) and vasopressor (noradrenaline 0.17 μg/kg/min) support. Ventilator-associated pneumonia (Streptococcus pneumoniae and Haemophilus influenzae).	Recovery. At 3-month follow-up, echocardiography confirmed the complete myocardial recovery (LVEF 65%) and cardiac magnetic resonance imaging showed no signs of residual myocardial fibrosis.
Pascariello et al.	NR	no	clinical deterioration, cardiogenic shock with severe hypotension, tachycardia, oliguria, anaemia and desaturation	Recovery. Discharged after 3 weeks.
Gaudriot et al.	T2 sequences showed diffuse hyperintense myocardium but suffered from too many artefacts to be considered diagnostic quality. Steady-state free precession cine images demonstrated biventricular cardiomyopathy (LVEF: 25%) with myocardial wall thickening. First-pass contrast-enhanced CMR did not reveal a subendocardial perfusion defect. Late gadolinium enhancement images demonstrated massive, heterogeneous, and predominantly subepicardial enhancement of the LV myocardium indicating a severe inflammation without evidence for ischaemic heart disease	Explantated heart, with large areas of myocardial necrosis, suppurated lesions, lymphocytic infiltration, polymorphic inflammatory infiltrate, mostly in myocardium areas but focally extended to endocardium and pericardium. Lymphocytes were predominant. Infiltrate also contained plasmocytes, neutrophils, eosinophils, and histiocytic and giant cells. Clusters of neutrophils with leucocytoclasia suggested suppurated lesions. Myocardial cells suffered from clarified cytoplasm, sometimes fibrillary or eosinophil, and enlarged dystrophic nucleus. Large areas of necrosis and haemorrhage were seen among these inflammatory areas. The adjacent myocardium showed interstitial oedema and focally recent interstitial fibrosis. Ziehl, Grocott, Gram, and EBER ISH stains were negative. Most lymphocytes were T phenotype CD5+, with only a few associated B lymphocytes CD20+. Numerous histiocytic cells were revealed by CD163 immunostaining. Cytomegalovirus immunostaining was negative as well as Epstein–Barr virus in situ hybridization.	patient became hypotensive and hypoxaemic (nasal cannula, oxygen 4 L/min) with clinical signs of pulmonary oedema, rapidly progressive biventricular hypokinetic non-dilated cardiomyopathy, myocarditis fulminant with cardiogenic shock	The patient recovered and was discharged for cardiovascular rehabilitation without any persistent respiratory or cardiac symptoms.
Shah et al.	NR/normal biventricular volumes and function, akinesia of the apico-lateral, apico-inferior walls and apical cap, myocardial hyperemia, and myocardial scar/necrosis consistent with a diagnosis of myocarditis	NR	acute kidney injury, ARDS/NR	discharged on day 16/chest pain-free upon discharge
Shabbir et al.	MRI of both lower limbs confirmed features of diffuse myositis with symmetrical appearances involving the anterior, medial and posterior muscle compartments of the thighs with subcutaneous oedema. A cardiac MRI showed normal LV and RV function, with possible evidence of myocardial oedema in the basal inferoseptum, in keeping with myocarditis, as well as a 16 mm circumferential pericardial effusion	NR	myositis	discharged day 13
Gauchotte et al.	NR	Biopsy after death—Microscopic examination of the heart revealed the existence of a multifocal inflammatory infiltration in both ventricles and septum, composed in its majority of macrophages and lymphocytes, associated with a mild polymorphonuclear neutrophils infiltrate. The myocardium was edematous, containing dystrophic cardiomyocytes, without necrosis. A hypocellular confluent area of fibrosis was found in the posterior wall of LV. There was no significant inflammatory infiltrate in the wall of capillaries and vessels.	AKI KDIGO 3, need for vasoactive support, cardiogenic shock	deceased 6 days after admission
Bernal-Torres et al.	inflammatory manifestations, with the recovery of the ejection fraction	patient did not consent	cardiogenic shock	discharged day 16
Oleszak et al.	NR	NR	respiratory distress	discharged
Gay et al.	NR	NR	respiratory failure and shock, oliguric kidney failure	recovered from critical illness, awaiting discharge to a rehabilitation centre
Gomila-Grange et al.	NR	NR	cardiogenic shock	One month after discharge, he was completely recovered
Richard et al.	myocardial necrosis, fibrosis, and hyperemia, indicating myocarditis according to the Lake Louise criteria	NR	ARDS, ventricular tachycardia, cardiogenic shock due to myocarditis fulminant, acute oliguric renal failure	The Impella device was removed, dobutamine was weaned off, and the patient was extubated the next day.
Wenzel et al.	patient A—native T1 map showing prolonged T1 relaxation times in the posterior interventricular septum and corresponding late gadolinium enhancement image with enhancement in the posterior septum, consistent with acute myocarditis. patient B—T2-short TI inversion recovery image, showing diffuse myocardial oedema and late gadolinium enhancement image with a subtle subepicardial enhancement of the lateral wall.	Immunohistochemistry and histology revealed myocardial inflammation in the absence of cardiomyocyte necrosis (Dallas criteria of ‘borderline myocarditis’6), with increased lymphocytes (CD3, LFA-1, and CD45R0) and macrophages (Mac-1), in part with highly abundant perforin-positive cytotoxic T cells. The inflammatory process seemed to be paralleled by increased thickness of small arteries	NR	discharged
Bemtgen et al.	NR	Significant infiltration of immune cells. Especially CD68+ macrophages and CD3+ T cells were found to be located primarily around small vessels within the myocardium, as shown by immunohistochemical stainings. Masson Trichrome and HE stainings further demonstrated the presence of perivascular fibrosis in serial tissue sections, but no myocyte necrosis	septic shock, severe end-organ failure	recovery
Beşler et al.	LV function, volumes, and mass were in normal range (LVEF: 64%, SV: 62.2 mL, LVEDV: 97 mL, LVESV: 34.8 mL, mass 128 g), T2 short tau inversion recovery sequence showed a subepicardial high-intensity signal in the mid posterolateral wall of LV which suggests myocardial wall oedema, subepicardial late gadolinium enhancement of the posterolateral wall in the mid-ventricle suggestive of myocarditis at 5 and 10 min after contrast administration, respectively	no	no	recovery
Meel et al.	normal LV and RV function and wall thickness with an LVEF: 65%, RVEF: 56% and wall thickness at end-diastole of 7 mm, delayed late gadolinium enhancement within the mid-wall as well as the epicardial regions involving the LV basal inferolateral wall, mid-anterolateral and mid-inferolateral wall T2 short tau inversion recovery black blood, high signal intensity within the LV anterolateral, inferolateral wall at the base and mid-ventricle level, on non-contrast T1 mapping, there was a prolongation of T1 time at the mid-ventricle segment 5 (1212 ± 141 ms) and segment 6 (1113 ± 93.8 ms)	no	no	recovery
Paul et al.	late subepicardial enhancement predominating in the inferior and lateral wall	no	no	recovery
Coyle et al.	diffuse biventricular and biatrial oedema with a small area of late gadolinium enhancement	no	cardiogenic shock, severe ARDS,	recovery
Irabien-Ortiz et al.	NR	no	cardiac arrest—electrical activity with no pulse and requiring cardiopulmonary resuscitation, emergent pericardiocentesis (drainage of serous fluid), and high-dose vasopressors for hemodynamic recovery	recovery
Frustaci et al.	confirmed normal LV dimensions and function (LVEF: 59%) and revealed the presence of right and left apical microaneurysms, late gadolinium enhancement imaging showed a subepicardial area in the inferolateral wall on basal and midventricular planes; corresponding mapping sequences documented a focal increase in native T1 values, extracellular volume fraction and T2 mapping values consistent with diffuse oedema and combined extracellular matrix expansion	diffusely mononuclear infiltrates associated with necrosis of adjacent myocytes and necrotizing vasculitis of intramural coronary arteries associated to positivity of CD45RO+ T-lymphocytes of the affected vessel wall. Arterioles not infiltrated by inflammatory cells presented at immunohistochemistry complement fractions (C3d) deposition suggesting coexisting endothelitis. Moreover, real-time PCR on two frozen endomyocardial samples for SARS-CoV-2 and the most common cardiotropic viruses, including adenovirus, cytomegalovirus, parvovirus B19, Epstein–Barr virus, human herpes virus 6, and herpes simplex virus 1 and 2, enterovirus, influenza virus A H1N1 and B and hepatitis C virus were negative	no	recovery
Sardari et al.	normal LV size with mildly reduced LVEF: 50%; T2-weighted oedema sequence with its post-analysis T2 ratio showed oedema/inflammation in the mid inferoseptal and inferior wall. Late gadolinium enhancement showed subepicardial fibrosis in the mid-inferior wall	not performed	no	NR
Warchoł et al.	LA enlargement and global left ventricular hypokinesia LVEF: 20%, T2-weighted sequence did not show myocardial oedema, late gadolinium enhancement with a large, patchy, and linear nonischemic pattern of fibrosis localized subepicardially and intramurally in the basal and mid cavity segments of the inferior and inferolateral wall and the apical segments of the inferior wall	not performed	NR	NR
Shahrami et al.	NR	not performed	a combination of cardiogenic and septic shock, respiratory arrest	death
Gaine et al.	biventricular dysfunction with an LVEF of 30%, STIR sequences showed biventricular oedema and reduced myocardial T1	not performed	NR	recovery
Taouihar et al.	focal hypertrophy in the anteroseptal and inferior segment of the LV extended over approximately 62 mm, and in anteroinferior of the RV extended over 27 mm, in hyper signal T2, with an intense enhancement on the tardive injected sequences compared to the rest of the myocardium in favour of a myocarditis	not performed	NR	discharged on day 7
Dahl et al..	diffuse myocardial oedema	not performed	respiratory distress with oliguria and hypotension, facial nerve palsy after discharge	discharged 11 days after admission
Matsumura et al..	high signal of T2-weighted black blood in anterior, interventricular septal, and posterolateral walls of the LV, late gadolinium enhancement in the posterolateral wall of the LV	infiltration of interstitial mononuclear cells Infiltrating cells included: T-cells and macrophages, more CD4-positive cells than CD8-positive cells, and few B-cells. RT-PCR SARS-CoV-2 negative in the myocardium. Electron microscopy showed inflammatory cell infiltration and myocyte damage compatible with myocarditis. Viral particles were not found in the high-power fields. Tests for antibodies against adenovirus, coxsackie virus, echovirus, and parainfluenza virus were negative.	Ten days after admission, bilateral peripheral facial nerve palsy occurred;	discharged 24 days after admission
Praet et al..	minimal pericardial effusion and contrast enhancement in pericardium but not in myocardium	Mononuclear infiltration of the myocardium, predominantly with T lymphocytes and macrophages, Myocardial biopsy shows mononuclear infiltration of the myocardium, with CD3-positive lymphocytes (arrow) and CD68-positive macrophages (arrow), without formation of granulomas	NR	recovered
Masiak et al.	NR	NR	Kikuchi-Fujimoto disease	discharged, at 7 months, physical impairment persists, and on echocardiography reduced LVEF: 53% with impaired global systolic heart function is still present
Noori et al.	mild to moderately reduced LVEF: 38%; moderate hypokinesis of midventricular inferolateral wall; oedema/inflammation of epicardium and epicardial scar of basal anteroseptal and anterior wall and midventricular anterolateral, inferolateral and inferior wall	not performed	no	discharged
Okor et al.	NR	NR	acute on chronic respiratory failure	death
Bulbul et al.	NR	NR	cardiogenic shock secondary to severe myocarditis	recovery
Khatri et al.	NR	NR	gastrointestinal bleeding and multiorgan failure	death
Nicol et al.	normal left ventricular size (LVEDV index: 75 mL/m^2^) and mild systolic dysfunction (LVEF: 45%) with global hypokinesia, presence of myocardial inflammation was confirmed by T2 mapping (global T2 relaxation times: 62 ms; centre-specific cut-off value for acute myocarditis: ≥55 ms), late gadolinium enhancement imaging (inversion time by using the Look-Locker technique: 280 ms) showed focal lateral subepicardial enhancement with prolonged T1 relaxation times (global T1 relaxation times: 1160 ms; centre-specific cut-off value for acute myocarditis: ≥1000 ms), small pericardial effusion,	multiple foci of lymphocytes in a diffuse inflammatory and oedematous background.	NR	recovery
Hudowenz et al.	late gadolinium enhancement of the entire LV myocardium with intracardial thrombi, T1 and T2 times were markedly prolonged, reflecting acute oedema following myocardial inflammation	active lymphocytic myocarditis	acute renal injury with microhaematuria	recovery
Jacobs et al.	NR	Slightly diffuse interstitial mononuclear inflammatory infiltrates, dominated by lymphocytes. No thrombotic events were observed in the microcirculation of the heart. In the patchy areas, the lymphocytes’ interlocked’ the myocytes, resulting in myocyte degeneration and necrosis (piecemeal necrosis). Intermingled with lymphocytes, only a few individual polymorphous neutrophils were found in the affected areas. Signs of inflammation were present in both the epicardium and the myocardium.	cardiogenic shock, ARDS	death
Hussain et al.	NR	not performed	ARDS	NR
Rehman et al.	not performed	not performed	no	NR
Monmeneu et al.	Right pulmonary condensation along with associated pleural effusion, LV of normal volume with concentric hypertrophy and mild depressed systolic function (LVEF: 53%) secondary to diffuse hypokinesia, global longitudinal strain was decreased. Myocardial oedema with a predominant subepicardial pattern was seen in the lateral, anterior, inferior, and apical segments on T2 short tau inversion recovery and T2 mapping sequences, determining wall pseudo-hypertrophy. Pericardial oedema without associated effusion was also observed. Native T1 and extracellular volume were increased in relation to the acute inflammatory process. Late gadolinium sequences showed extensive, patchy intramyocardial/subepicardial enhancement affecting the entire lateral, anterior, inferior, and apical septal walls and the pericardium.	NR	no	recovery
Tavazzi et al.	NR	Low-grade interstitial and endocardial inflammation. Large, vacuolated, CD-68-positive macrophages were seen with immunelight microscopy. The ultrastructural study demonstrated single or small groups of viral particles with the morphology and size of coronaviruses. The viral particles were observed in cytopathic, structurally damaged interstitial cells that demonstrated loss of cytoplasmic membrane integrity. Cardiac myocytes showed non-specific features of local myofibrillar lysis and lipid droplets. Interstitial fibrosis was minimal, focal, and mainly perivascular.	cardiogenic shock, sepsis	death

AKI—acute kidney injury, ARDS—acute respiratory distress syndrome, CAD—coronary artery disease, CMR—cardiac magnetic resonansce, CPR—cardiopulmonary resuscitation, DIC—disseminated intravascular coagulation, DKA—diabetic ketoacidosis, DSA—digital subtraction angiography, ECMO—extracorporeal membrane oxygenation, EMB—endomyocardial biopsy, HF—heart failure, hs-TnT—high sensitive troponins, IV—intravenous, IVS—interventricular septum thickness, LA—left atrium, LAED—left atrial end-diastolic diameter, LV—left ventricle, LVDD—left ventricle end-diastolic dimension, LVDS—left ventricle end-systolic dimension, LVED—left ventricular end-diastolic diameter, LVEF—left ventricular ejection fraction, LVH—left ventricular hypertrophy, LVIDd—left ventricular internal diameter during end diastole, LVIDs—left ventricular internal diameter during end systole, LVPW—left ventricular posterior wall, MCS—mechanical cardiac support, MRI—magnetic resonance imaging, MVR—mitral valve regurgitation, NR—not reported, NSTEMI—non-ST-segment elevation myocardial infarction, NTproBNP—N-terminal-pro B-type natriuretic peptide, PAP—pulmonary artery pressure, PASP—pulmonary artery systolic pressure, PCR—polymerase chain reaction, PE—pulmonary embolism, RAED—right atrial end-diastolic diameter, RT-PCR—real time reverse transcription-polymerase chain reaction, RV—right ventricle, RVED—right ventricular end-diastolic diameter, STiR—short-TI inversion recovery, TVR—tricuspid valve regurgitation, UNL—upper normal limit, VT—ventricular tachycardia.

## Data Availability

Full report of screened papers is available in Appendix A.

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
