# Peer review of "COVID-19 Related Myocarditis in Adults: A Systematic Review of Case Reports"

_jcm, 2022, doi:10.3390/jcm11195519_

Round 1

Reviewer 1 Report

In the present manuscript, the authors performed a systematic review of case reports on covid-19 related myocarditis. The methods are well described and scientifically sound. The results well summarizes the epidemiology and clinical features of this potentially life threatening complcations of Sars-Cov2 infection.

I just have some minor comments pertaining data presentation.

- The amount of results is remarkably large, and an additional figure (more illustrations/less words) could be used to portray the study results (i.e. a forest plot)

- Conclusions could be brieafer, with no references, summarizing take home messages within just a couple of sentences

Reviewer 2 Report

This article thoroughly reviewed the published case reports regarding COVID-19 related myocarditis. According to 59 studies and 70 patients, the most common presentation were middle-aged men with shortness of breath. Two-thirds were diagnosed with myocarditis during COVID-19 infection, while one-third were diagnosed with myocarditis followed infection after a mean time of 51 days. Endomyocardial biopsies were done in one-third of cases, but with the help of multimodal imaging including cardiac MRI and echocardiography, cases could be confirmed. Therapeutic regimens varied significantly.

The topic is interesting and timely, but at some point, it's inadequately presented by an unsuccinct way.

1. Tables 1, 2, and supplementary table contain too much information and are not readable because they are too long

2. Even though the pathophysiology of COVID-19 related myocarditis is not well known, it would be better to discuss in detail including other hypotheses (e.g. ACE2) other than cytokine storm.

3. Basically this article is based on the published case reports, novelty is doubtful. Center for Disease Control and Prevention already reported that myocarditis could be diagnosed several months after COVID-19 infection although this report used International Classification of Diseases, Tenth Revision, Clinical Modification. (MMWR Morb Mortal Wkly Rep. 2021 Sep 3;70(35):1228-1232)

Round 2

Reviewer 2 Report

This article summarized current evidence on the COVID-19 related myocarditis in details and discussed relevant issues. Central illustration and harvest plot are clear and informative.